# The Efficacy of Polydioxanone Sutures in Treating Mild-to-Moderate Knee Osteoarthritis: A Systematic Review and Meta-Analysis

**DOI:** 10.3390/medicina61030388

**Published:** 2025-02-24

**Authors:** Eun-Ju Lee, Hyoung-Ye Kim, Dong-Woo Lee, Sai-Won Kwon

**Affiliations:** 1Department of Public Health, Graduate School, The Catholic University of Korea, 222 Banpo-daero, Seocho-gu, Seoul 06591, Republic of Korea; junestyle98@naver.com; 2Department of Orthopaedic Surgery, Soonchunhyang University Cheonan Hospital, 31, Suncheonhyang 6-gil, Dongam-gu, Cheonan 31151, Republic of Korea; hyoungnoo@gmail.com (H.-Y.K.); acn8513@gmail.com (D.-W.L.)

**Keywords:** knee osteoarthritis, polydioxanone, hyaluronic acid, sodium polynucleotide, 100 mm visual analogue scale, meta-analysis

## Abstract

*Background and Objectives*: This meta-analysis aimed to compare the efficacy of polydioxanone (PDO) suture, a non-surgical treatment for knee osteoarthritis, with intra-articular hyaluronic acid (HA) injections. *Materials and Methods*: A comprehensive literature search was conducted using major databases including MEDLINE, EMBASE, Cochrane Library, KoreaMed, KMBASE, and RISS. Randomized controlled trials (RCTs) published up to 30 April 2024, focusing on knee osteoarthritis, pain, PDO suture, and intra-articular injections, were included. A total of 10 RCTs were analyzed, with participants having Kellgren & Lawrence Grade II-III knee osteoarthritis. This study compared the pain relief effects of PDO suture and HA injections. *Results*: The meta-analysis results showed that PDO suture demonstrated consistent and significant pain reduction over a 30-week observation period (*p* < 0.05), while HA injections did not exhibit statistically significant pain relief. *Conclusions*: PDO sutures offer the potential for long-term pain management in patients with knee osteoarthritis. However, this study has limitations such as the heterogeneity among studies, and given that the efficacy of PDO sutures is based on a single study, further research is needed to establish the long-term safety profile of polydioxanone sutures and to ensure the generalizability of the findings.

## 1. Introduction

Osteoarthritis (OA) is a degenerative joint disease primarily associated with aging and is characterized by decreased osteoblast differentiation and cartilage damage [1,2]. While more common in the elderly, studies have shown OA can also affect the younger population, making it a significant cause of disability globally [2]. The development of osteoarthritis (OA) is influenced by a variety of factors, including obesity, physical activity, genetic predispositions, previous joint injuries, work-related conditions, and anatomical abnormalities in the joints [2,3]. The knee connects the femur and tibia, the two longest levers in the body, which increases its susceptibility to degenerative changes [4,5,6]. In knee OA, quadriceps muscle weakening reduces joint stability, leading to muscle atrophy and exacerbating pain, functional impairment, and reduced quality of life [7,8,9,10,11].

OA treatment aims to alleviate pain, stiffness, and dysfunction while preventing deformity and impaired mobility [6]. Treatment options include non-surgical approaches such as weight loss, muscle strengthening, physical therapy, and taping, which have shown effectiveness in managing symptoms [12,13,14,15,16,17,18]. When conservative methods fail, pharmacological treatments like NSAIDs or intra-articular injections (e.g., HA, PN) are used [19]. These injections offer localized relief and have demonstrated efficacy in pain reduction, with HA showing benefits up to 12 months [20,21,22,23,24,25,26]. Polydioxanone (PDO) has been used to treat musculoskeletal pain and has shown promise in combination with physical therapy for conditions like chronic neck pain [27]. A study on knee OA patients demonstrated that PDO sutures effectively relieved knee pain by strengthening the vastus medialis muscle [28]. This study aims to compare the pain-relieving effects of PDO, HA, and PN injections in patients with knee OA (Kellgren & Lawrence Grade II-III), contributing to a better understanding of PDO as a non-surgical treatment option.

## 2. Materials and Methods

### 2.1. Search Strategy

This meta-analysis was performed in accordance with the Preferred Reporting Items for Systematic Reviews and Meta-Analyses (PRISMA) guideline (Appendix A). The study protocol was registered in a prospective registry of systematic reviews (CRD number: 42024621007). The study involved a comprehensive literature search to investigate interventions influencing pain in knee osteoarthritis patients. We searched both domestic and international academic databases, including MEDLINE, EMBASE, Cochrane Library, KoreaMed, KMBASE, and RISS. Additional data were obtained through manual searches of Google Scholar and reference lists. The final date for article selection was 30 April 2024, and there was no publication date restriction. The literature search was conducted according to the PICO (Patient, Intervention, Comparison, Outcome) criteria [29]. The patients in this study are individuals with knee osteoarthritis. The search terms for knee osteoarthritis included “Knee osteoarthritis”, “Knee arthritis”, “Osteoarthrosis”, “Gonoarthrosis”, and “Gonoarthritis”. The intervention was restricted to medical devices applied in the treatment of knee osteoarthritis, and the study investigated intra-articular hyaluronic acid injections, intra-articular polynucleotide injections, and polydioxanone sutures. The search terms for intra-articular hyaluronic acid injection were “Hyaluronate acid” or “HA” or “Hyaluronate” or “Hyaluronan” or “Viscosupplementation”, and for intra-articular polynucleotide injections, they were “Polynucleotide” or “PN”. The search term for polydioxanone sutures was “Polydioxanone”. The comparison was searched using “Sham” or “Placebo” [30]. The outcome values were measured using the following scale—the 100 mm visual analogue scale (100 mm VAS) [21,31,32].

### 2.2. Study Selection and Data Extraction

Two independent reviewers assessed studies for inclusion based on pre-defined eligibility criteria. Titles and abstracts were screened for relevance, with full-text assessment conducted for uncertain cases. Disagreements were resolved through discussion. Studies meeting the PICO criteria were included [29]. (1) Population: patients diagnosed with moderate-to-severe knee osteoarthritis (Kellgren & Lawrence grade II–III on X-ray) and experiencing pain in one or both knees of 40 mm or more on weight-bearing pain assessment. Patients with rheumatoid arthritis or other inflammatory arthritis that could affect study results were excluded [33]. (2) Intervention: intra-articular hyaluronic acid injections, intra-articular polynucleotide injections, and polydioxanone sutures. (3) Comparison: sham or placebo. (4) Outcome: 100 mm VAS. (5) Study design: randomized controlled trials (RCTs) and systematic reviews. Case reports, non-randomized studies, and studies with inaccessible full texts were excluded. The publication language was restricted to English and Korean. The study design was performed according to Cochrane Review Methods. All data presented in this study were extracted from published articles, and no personal data were included; therefore, ethical approval was not required.

### 2.3. Quality Assessment

Two independent reviewers assessed the risk of bias in the RCTs using the Cochrane Collaboration’s Risk of Bias tool. The assessment was based on seven domains: random sequence generation (selection bias), allocation concealment (selection bias), blinding of participants and personnel (performance bias), blinding of outcome assessment (detection bias), incomplete outcome data (attrition bias), selective reporting (reporting bias), and other bias. Each domain was assessed as having a high, low, or unclear risk of bias. Any disagreements between the two reviewers were resolved through discussion until consensus was reached [34].

### 2.4. Outcome Measure and Time Points

In clinical trials involving knee osteoarthritis patients, pain was set as the primary outcome measure for this study. The average change in pain scores was assessed using the 100 mm VAS compared to baseline during weight bearing. Since treatment duration and follow-up evaluation points varied among studies, we categorized the results of individual studies into seven intervals: 1 week (1–3 weeks), 4 weeks (4–7 weeks), 8 weeks (8–11 weeks), 12 weeks (12–15 weeks), 16 weeks (16–19 weeks), 20 weeks (20–23 weeks), and 24 weeks (24 weeks or more). This approach was designed to best capture the data presented in all studies.

### 2.5. Statistical Analysis

All statistical analyses were conducted using the Review Manager software (Revman Version 5.3: The Cochrane Collaboration, Copenhagen, Denmark). Data were pooled and analyzed using a fixed-effects model; if heterogeneity was detected, a random-effects model was employed for the meta-analysis. Effect estimates for medical outcomes were described as the mean difference (MD) if one outcome measure was the same, as the standardized mean difference (SMD) if all outcome measures were the same, and as the SMD if the outcome measures differed. Heterogeneity was assessed using the I^2^ test, with a value exceeding 50% indicating substantial heterogeneity. Publication bias was evaluated for the meta-analysis using funnel plots, checking for symmetry of the effect sizes on both sides of the null hypothesis to determine the presence of publication bias [35].

## 3. Results

### 3.1. Study Characteristics

The literature selection and exclusion criteria for this study are illustrated in the flowchart (Figure 1) [36]. An initial search identified 27 studies, and an additional 6 studies were included through manual searching. After reviewing titles, abstracts, and full-text articles, a total of 10 studies were ultimately included in the meta-analysis. Details of the selected studies are reflected in Table 1.

### 3.2. Risk of Bias Assessment

The Cochrane Collaboration’s Risk of Bias (ROB) tool was applied to assess the risk of bias in each domain for the 10 selected randomized controlled trials. The results of the risk of bias assessment were visualized using the RevMan software. All 10 studies reported that appropriate randomization sequences were generated. Allocation concealment was adequately described in four studies. Blinding of participants and personnel was clearly performed in six studies, while one study did not mention blinding and was therefore assessed as having a high risk. Blinding of outcome assessment was performed in 9 out of 10 studies. The risk of selective reporting bias was low (Appendix A).

### 3.3. 100 mm VAS

All 10 studies were written in English and were RCTs conducted domestically and internationally between 1998 and 2022. A total of 1933 patients were included in these studies. Of these, 1006 patients were in the intervention group: 976 patients received HA injections and 30 patients received PDO sutures. PN injections were excluded from the final analysis due to a lack of studies comparing them to a control group. The control group consisted of 927 patients. The mean age of patients in each study ranged from 53 to 72 years. All 10 studies showed a higher participation rate of female patients compared to male patients. The mean baseline 100 mm VAS score ranged from approximately 46 mm to 71 mm.

Among the 10 studies collected, 5 studies investigating the 1–3 week time point after intra-articular HA injections showed that HA injections (*n* = 699) resulted in an average reduction of 3.13 mm in the 100 mm VAS compared to the control group (*n* = 676) (95% CI: −5.04 to −1.23 mm, *p* = 0.001), which was statistically significant (Figure 2A). At the 4–7 week time point, six studies reported that HA injections (*n* = 726) reduced the average 100 mm VAS by 3.61 mm compared to the control group (*n* = 734), but this was not statistically significant (95% CI: −7.35 to 0.12 mm, *p* = 0.06). In contrast, a single study on PDO sutures found that PDO sutures (*n* = 30) resulted in a significant average reduction of 22.4 mm in the 100 mm VAS compared to the control group (*n* = 10) (95% CI: −38.04 to −6.76 mm, *p* = 0.005) (Figure 2B). At the 8–11 week time point, four studies showed that HA injections (*n* = 330) led to an average reduction of 6.12 mm in the 100 mm VAS compared to the control group (*n* = 295), but this was not statistically significant (95% CI: −17.68 to 5.45 mm, *p* = 0.30). However, a single study on PDO sutures found a significant reduction of 21.3 mm in the 100 mm VAS with PDO sutures (*n* = 30) compared to the control group (*n* = 10) (95% CI: −36.74 to −5.86 mm, *p* = 0.007) (Figure 2C). At the 12–15 week time point, seven studies reported that HA injections (*n* = 876) resulted in an average reduction of 1.59 mm in the 100 mm VAS compared to the control group (*n* = 810), but this was not statistically significant (95% CI: −3.65 to 0.46 mm, *p* = 0.13) (Figure 2D). At the 16–19 week time point, three studies showed that HA injections (*n* = 487) reduced the average 100 mm VAS by 5.35 mm compared to the control group (*n* = 494), but this difference was not statistically significant (95% CI: −13.35 to 2.65 mm, *p* = 0.19) (Figure 2E). At the 20–23 week time point, two studies indicated that HA injections (*n* = 181) resulted in an average reduction of 2.7 mm in the 100 mm VAS compared to the control group (*n* = 171), but this was not statistically significant (95% CI: −7.89 to 2.48 mm, *p* = 0.31). In contrast, a single study on PDO sutures found a significant reduction of 18.4 mm in the 100 mm VAS with PDO sutures (*n* = 30) compared to the control group (*n* = 10) (95% CI: −33.8 to −3.0 mm, *p* = 0.02) (Figure 2F). At the 24–30 week time point, seven studies showed that HA injections (*n* = 890) led to an average reduction of 3.52 mm in the 100 mm VAS compared to the control group (*n* = 840), but this difference was not statistically significant (95% CI: −7.28 to 0.24 mm, *p* = 0.07). Conversely, a single study on PDO sutures reported a significant reduction of 17.13 mm in the 100 mm VAS with PDO sutures (*n* = 30) compared to the control group (*n* = 10) (95% CI: −32.58 to −1.68 mm, *p* = 0.03) (Figure 2G).

## 4. Discussion

In this meta-analysis, the average 100 mm VAS scores at the 4–7 week, 8–11 week, 12–15 week, 16–19 week, 20–23 week, and 24–30 week time points were reduced by 3.61 mm, 6.12 mm, 1.59 mm, 5.35 mm, 2.7 mm, and 3.52 mm, respectively. However, these reductions were not statistically significant. This suggests that while intra-articular HA injections may contribute to alleviating knee osteoarthritis pain to some extent, the effect is not consistent and has not been statistically proven. The clinical significance of the pain reduction effect of HA injections may be limited. Additionally, the small sample sizes and differences in study designs among the included studies may have contributed to the high heterogeneity and potentially impacted the meta-analysis results.

Hyaluronic acid plays a crucial role in joint lubrication and shock absorption, enabling smooth joint movement. While the viscoelastic properties of normal joints are maintained by HA, in patients with knee osteoarthritis, HA is degraded due to inflammatory responses, leading to reduced synovial fluid viscosity and resulting in joint pain [12,46]. Colen et al. (2012) conducted a meta-analysis on intra-articular HA injections and found a reduction of 10.2 mm (95% CI −15.97, −4.42) on the 100 mm VAS after 3 months compared to placebo, indicating pain reduction [47]. The Korean Health and Medical Technology Assessment Institute (NECA) report on “Intra-articular Drug Injections for Knee Osteoarthritis” found that HA injections reduced pain compared to placebo for up to 12 months, although no significant difference was observed beyond 12 months [26]. Intra-articular HA injections may be considered for patients with moderate or less severe osteoarthritis who cannot use NSAIDs, with pain reduction effects typically observed within 14 weeks. However, a 2006 Cochrane review analyzing 76 studies reported significant pain relief and functional improvement within the initial 4 weeks but noted variability between studies and publication bias [46]. Recent guidelines from the American Academy of Orthopaedic Surgeons (AAOS) and the American College of Rheumatology (ACR) have seen a reduction in the recommendation level due to ongoing debate about the superiority of HA over placebo [48,49]. Although not analyzed in this meta-analysis, there are clinical results from some studies suggesting that intra-articular PN injections may provide similar or even more significant pain relief compared to intra-articular HA injections [50,51]. However, given that PN injections were approved in Korea only in 2017, there are fewer clinical studies compared to other treatments for knee osteoarthritis, necessitating larger-scale clinical research for a more accurate assessment [52].

In the treatment of knee osteoarthritis, muscles are closely related to joint pressure. A prolonged recovery period with limited muscle use can lead to muscle atrophy and a decrease in muscle mass, size, and strength [6,53]. In particular, the quadriceps play a crucial role in absorbing shocks applied to the joint and reducing the load, thereby stabilizing the lower limb. Weakening of the quadriceps reduces joint stability and shock absorption, and the lack of load on the joint ultimately leads to atrophy of unused muscles [53,54]. In these patients, strengthening and supporting the vastus medialis can help alleviate pain by reducing the load on the knee joint. Unlike conventional treatments that primarily rely on intra-articular injections, polydioxanone sutures are directly inserted into the vastus medialis of a leg to provide stabilization and support. Through this technique, the sutures compensate for the muscle contraction force necessary for activity, ultimately reducing joint load and alleviating pain [28,55]. Additionally, polydioxanone sutures induce micro-injuries in the muscles, triggering an initial inflammatory response and promoting muscle regeneration. Animal studies confirmed an increase in inflammation markers and muscle regeneration markers, leading to myofiber hypertrophy and myogenesis. Moreover, activation of muscle stem cells plays a crucial role in muscle recovery. This treatment has also been shown to improve muscle strength in both clinical and animal trials [56,57]. The meta-analysis results for polydioxanone sutures showed consistent and significant pain reduction across the study period. Compared to placebo, the mean 100 mm VAS decreased by 22.4 mm, 21.3 mm, and 18.4 mm at the 1–3 week, 4–7 week, and 8–11 week time points, respectively. Furthermore, at 12–15 weeks, 16–19 weeks, 20–23 weeks, and 24–30 weeks, the mean 100 mm VAS decreased by 21.3 mm, 18.4 mm, 17.13 mm, respectively, compared to placebo, all falling within 10–30 mm, which is considered the Minimal Clinically Important Difference (MCID) in pain studies [58,59,60]. These findings suggest that polydioxanone sutures may contribute to improving the quality of life for patients with knee osteoarthritis by alleviating pain. Additionally, the intra-articular HA injections were observed up to 26 weeks in this meta-analysis, whereas polydioxanone sutures showed pain relief effects up to 30 weeks, suggesting the potential for long-term efficacy in reducing pain in patients with knee osteoarthritis. However, aside from the differences in observation periods between the two treatments, various factors such as patient characteristics and the number of applications could influence the outcomes. Therefore, further research is necessary for a more detailed comparative analysis. Moreover, future studies should aim to provide clearer evidence of the long-term efficacy and safety of polydioxanone sutures. Additionally, a limitation of this meta-analysis is the lack of available data from the included studies. Due to the variation in treatment durations and post-treatment evaluation time points across different studies, it was not possible to compare all study data at specific time points. Therefore, the evaluation time points were grouped into intervals, and a random-effects model was included to control for heterogeneity. Safety data were not assessed in this study.

## 5. Conclusions

A meta-analysis of patients with moderate-to-severe knee osteoarthritis (Kellgren & Lawrence Grade II–III) found that the implantation of polydioxanone sutures into the vastus medialis significantly reduced knee pain over a 30-week follow-up period. These findings suggest that polydioxanone sutures may offer a promising long-term treatment option for pain management in knee osteoarthritis. However, the result of the efficacy of PDO sutures is based on a single study; therefore, further studies are required to establish the long-term safety profile of polydioxanone sutures and assess their effectiveness across a broader patient population to ensure the generalizability of the findings.

## Figures and Tables

**Figure 1 medicina-61-00388-f001:**
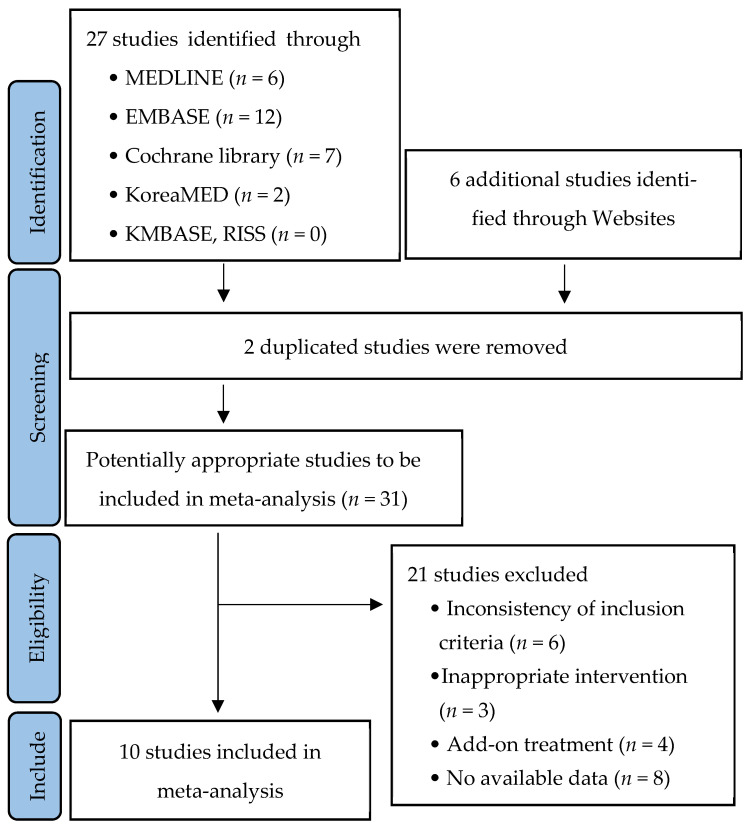
Summary of study population.

**Figure 2 medicina-61-00388-f002:**
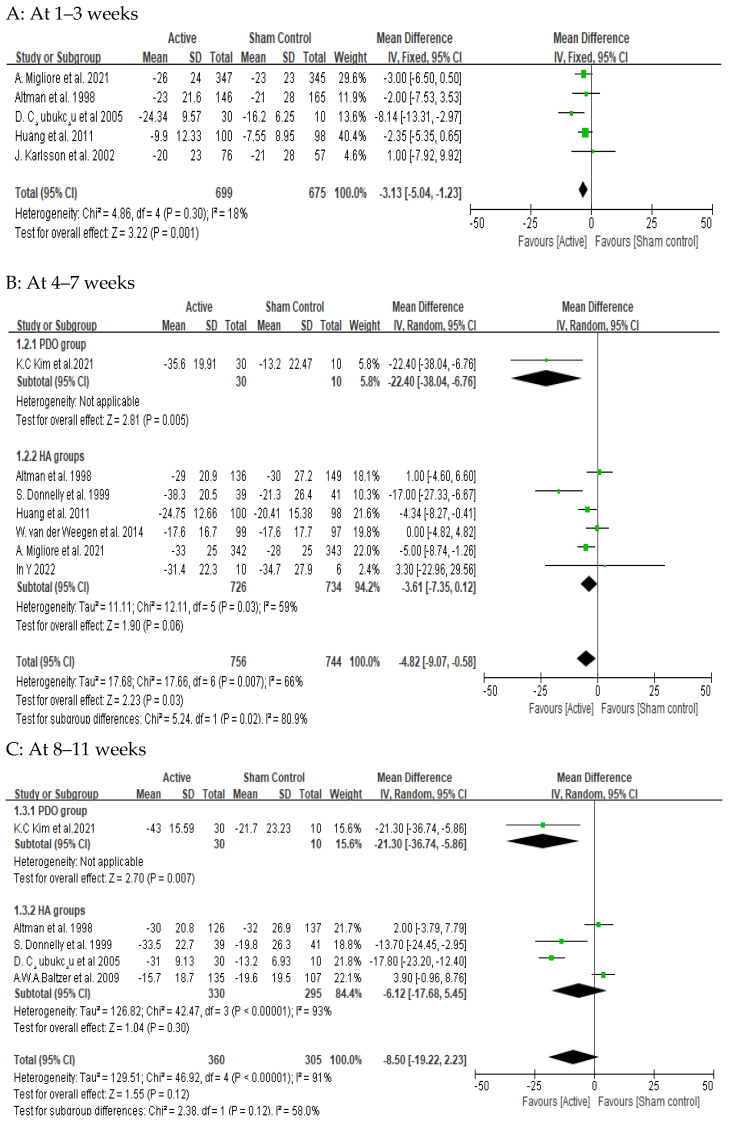
Forest plot of comparisons: Active group versus placebo for pain at each visit. Data are presented as weighted mean difference for each study (boxes), 95% CIs (horizontal lines), and summary weighted mean difference of 100 mm visual analogue scale (VAS) score with 95% CIs (diamond) [28,37,38,39,40,41,42,43,44,45].

**Table 1 medicina-61-00388-t001:** Major characteristics of included studies.

Study	Intervention	Assessment (Week)	Treatment Group			Placebo Group		
			Age (yr)	Gender (M/F)	100 mm VAS	Age (yr)	Gender (M/F)	100 mm VAS
Altman et al. [37]	HA: Hyalgan^®^	3, 4, 5, 9, 12, 16, 21, 26	62.0 ± 10.0	64/41	54.0 ± 29.0	65.0 ± 10.0	61/54	55.0 ± 29.0
S. donnelly et al. [38]	HA: Hyalgan^®^	5, 8, 16, 24	65.8 ± 8.8	12/38	65.8 ± 18.0	64.8 ± 9.3	21/29	61.9 ± 22.9
J. Karlsson et al. [39]	HA: Artzal^®^	1, 2, 3, 12, 20, 26	72.0 ± 7.0	22/54	64.0 ± 15.0	71.0 ± 6.0	20/37	65.0 ± 15.0
D. çubukcu et al. [40]	HA: Synvisc^®^	1, 2, 3, 8	52.6 ± 7.2	6/14	71.0 ± 1.2	57.6 ± 2.8	0/10	67.0 ± 2.1
A.W.A. Baltzer et al. [41]	HA: HYA-Ject^®^	7, 13, 26	57.4 ± 12.0	74/61	68.3 ± 12.8	60.3 ± 10.7	68/39	66.3 ± 14.5
Huang et al. [42]	HA: Hyalan^®^	1, 5, 13, 25	65.9 ± 8.1	26/74	47.9 ± 10.8	64.2 ± 8.4	22/78	45.7 ± 10.4
W. van der Weegen et al. [43]	HA: Fermathron plus	4, 12, 24	58.7 ± 9.6	49/50	56.4 ± 16.7	60.1 ± 10.1	50/47	58.2 ± 17.7
K.C. Kim [28]	PDO suture: MEST-B2375	4, 8, 20, 30	62.9 ± 8.5	6/24	63.9 ± 14.9	64.0 ± 5.7	2/8	62.3 ± 10.7
A. Migliore [44]	HA: Sinovial^®^	1, 6, 12, 18, 24	63.7 ± 8.7	115/232	63.0 ± 13.0	63.8 ± 8.1	115/230	65.0 ± 14.0
Yong In [45]	HA: YYD302	4, 12	60.7 ± 6.9	1/11	59.8 ± 10.1	61.7 ± 12.5	1/6	64.0 ± 12.4

HA: hyaluronic acid; PDO: polydioxanone; M/F: male/female.

## Data Availability

All data from this study are available upon reasonable request to the corresponding author.

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
