# Peer review of "The Efficacy of Polydioxanone Sutures in Treating Mild-to-Moderate Knee Osteoarthritis: A Systematic Review and Meta-Analysis"

_medicina, 2025, doi:10.3390/medicina61030388_

Round 1

Reviewer 1 Report

Comments and Suggestions for Authors

The abstract mentions "consistent and significant pain reduction" for PDO, but the data come from a single study. This should be rephrased to reflect the limited evidence.

The introduction effectively outlines the significance of osteoarthritis and current treatment modalities. However, it would benefit from a more detailed rationale for comparing PDO sutures with HA injections specifically. The authors should explicitly state the existing gaps in the literature that their study aims to address, such as limited comparative analyses between PDO sutures and HA injections. The study objectives are clear but could be enhanced by specifying the hypothesis or expected outcomes.

The manuscript does not mention how missing data were handled. Provide more information on the intervention specifics, handling of missing data, and subgroup analyses to enhance reproducibility and depth.

The discussion should delve deeper into possible mechanisms by which PDO sutures provide sustained pain relief compared to HA injections.

Expanding on limitations.

While the manuscript touches on the potential of PDO sutures, a more thorough discussion of how these findings could influence clinical practice guidelines or patient management strategies is warranted.

Standardizing formatting and citations as journal requirements are necessary. 

Comments on the Quality of English Language

There are minor grammatical errors.

Reviewer 2 Report

Comments and Suggestions for Authors

General characteristics and evaluation of the reviewed article:

The article evaluates the effectiveness of polydioxanone (PDO) sutures as a non-surgical treatment for knee osteoarthritis compared to intra-articular hyaluronic acid (HA) injections. The meta-analysis includes 10 randomized controlled trials (RCTs) involving patients with Kellgren & Lawrence Grade II-III osteoarthritis. Results indicate that PDO sutures provide consistent and statistically significant pain reduction over a 30-week period, unlike HA injections, which failed to achieve similar outcomes.

The study is commendable for its comprehensive literature search, focus on RCTs, and demonstration of PDO sutures' potential as a long-term pain management option. However, limitations include heterogeneity among the included studies, lack of focus on long-term safety, and insufficient exploration of the mechanisms underlying the treatment's effectiveness. Furthermore, the observation period is relatively short, given the chronic nature of osteoarthritis.

Despite these limitations, the findings suggest PDO sutures could be a promising alternative for managing osteoarthritis-related pain. Further research is needed to assess their long-term efficacy, safety, and mechanisms of action, as well as to standardize study protocols for future analyses.

The article is interesting, addresses a timely and important topic and definitely fits the scope of the journal. It is written generally correctly and requires only minor corrections and additions before further processing and acceptance for publication. Below are my points and detailed comments.

Minor comments:

The introduction could benefit from including information on diseases that ultimately lead to osteoarthritis, as this would provide a broader audience with a clearer understanding of the issue's scope. Adding a brief description of osteoarthritis along with relevant literature is recommended to enhance context.

Expanding the opening paragraph to delve deeper into osteoarthritis would reinforce the introduction by emphasizing the significance of this condition. Factors such as occupation, sports participation, musculoskeletal injuries, obesity, and gender influence the prevalence of osteoarthritis. Discussing these aspects, supported by appropriate references, would create a stronger foundation for the topic. The following references are suggested for inclusion in this section:

https://doi.org/10.3390/healthcare12161648

DOI: 10.1056/NEJMcp1903768

References should be numbered in the order they appear in the text.

The studies analyzed differed in terms of protocols, patient characteristics and assessment time points, which affects the interpretation of results and limits their generalizability. Future analyses should consider standardizing methods for assessing therapeutic effects to reduce heterogeneity and improve interpretation of results. Please describe this better in the limitations of the paper AND future plans.

The paper lacks a detailed analysis of the side effects and long-term safety of PDO sutures. It is recommended that studies evaluating the safety of PDOs over a longer period of time should be conducted to comprehensively evaluate their usefulness in osteoarthrosis therapy. Please discuss this in the discussion and expand the plan for further research.

The authors do not elaborate on how exactly PDO sutures contribute to pain reduction, which limits understanding of the potential benefits of the therapy. The authors should consider describing the biomechanical and biological mechanisms that explain the effectiveness of PDO in reducing pain. Please expand on this in your discussion with the necessary references.

Only 30 patients received PDO therapy compared to a much larger number of patients receiving HA, which affects the statistical power of the analysis. Sample sizes in the PDO groups should be increased to provide more reliable data and compare them with HA therapies on a level playing field.

Future studies should take into account demographic variables and indicators that affect patients' response to treatment, such as BMI, physical activity and comorbidities.

I congratulate the authors on the interesting paper and wish them further success.
